# Economic events and the volatility of government bill rates

Chao Xiao, Yu Lou *, Jie Liu, Yuan Zhao, Yikang Tian

School of Finance, Southwestern University of Finance and Economics, Chengdu, Sichuan, China

* louyu@smail.swufe.edu.cn

**Data Availability Statement:** The US data comes from Robert J. Barro's website (https://scholar.harvard.edu/barro/publications/macroeconomic-crises-1870-bpea). Other relevant data are within the paper and its Supporting Information file.

## Abstract

Many studies show that in many countries (especially the G7), volatility in government bill rates far exceeds that in consumption growth rates. This volatility puzzle cannot be predicted by traditional disaster models, in which rare economic disasters are defined as a peak-to-trough percent fall in consumption (or real per capita GDP) by a high threshold ($\geq 10\%$). For this purpose, we extend the traditional definition of rare economic disasters and propose a novel asset pricing model that models both good and bad events. We define a bad (or good) event as a peak-to-trough absolute decline (or a trough-to-peak absolute rise) in consumption growth rates by a low threshold (<10%). Compared to traditional disaster models, our model contains three improvements. First, model good and bad events, not just bad ones (e.g., rare economic disasters). Second, the event's impact lasts for multiple periods rather than one period. Third, model non-rare economic events. We calibrate the parameters in our model to match the moments from U.S. asset return data. Simulation results indicate that the model can successfully predict the volatility of U.S. government bill rates higher than that of U.S. consumption growth rates.

## Introduction

Mehra and Prescott [1] find that the average annual stock return and risk-free rate in the United States from 1889 to 1978 were about 6.98% and 0.8%, respectively; that is, the average equity premium in the United States was as high as 6.18%. The early consumption-based capital asset pricing models cannot explain such a high equity premium with a rational risk aversion coefficient (no higher than 10) called the equity premium puzzle. Rietz [2] put forward rare economic disasters to account for the puzzle. Barro [3] and Barro and Ursúa [4] define rare economic disasters and report the statistical results of rare economic disasters. Barro [3] defines a rare economic disaster as a peak-to-trough percent fall in real per capita GDP by at least 15%. Barro and Ursúa [4] define a rare economic disaster as a peak-to-trough percent fall in consumption (or real per capita GDP) by at least 10%. Numerous asset pricing models [3–10] built on the above definition of rare economic disasters are generally referred to as the Rietz-Barro model, which has occupied an important place in explaining the equity premium puzzle, the risk-free rate puzzle [11], the stock market volatility puzzle [12].

However, the Rietz-Barro model has faced the following criticisms: 1) Gourio [13] points out that the growth component of consumption has received insufficient attention; 2)

**Funding:** The authors received no specific funding for this work.

**Competing interests:** The authors have declared that no competing interests exist.

Nakamura et al. [14] indicate that rare disasters are modeled as a jump with no ongoing effects. In response to these criticisms, Nakamura et al. [14] allow for partial recoveries after disasters that unfold over multiple years. Barro and Jin [15] expand on the model of Nakamura et al. [14] by incorporating long-run risks and argue that rare disasters and long-run risks are complementary approaches to understanding asset pricing.

This paper intends to point out another shortcoming of the Rietz-Barro model. The Rietz-Barro model predicts that the government bill rate volatility is less than the consumption growth rate volatility; however, numerous studies show that the former far exceeds the latter in the data [4, 7, 15, 16]. For example, based on the data from Barro and Ursúa [4], the government bill rate volatility of the United States, the United Kingdom, Canada, Japan, Australia, France, and Germany is 4.82%, 6.24%, 11.99%, 14.75%, 5.66%, 9.96%, and 17.88% respectively. In comparison, the consumption growth rate of them is 3.60%, 2.83%, 4.74%, 6.89%, 5.06%, 6.74%, and 5.70% respectively. In other words, although the Rietz-Barro model can explain many puzzles in financial economics, it cannot explain this volatility puzzle. To explain the puzzle, we modify the Rietz-Barro model's definition of economic events in this paper.

In the economy, there are not only bad events represented by rare disasters but also good events represented by institutional reform, management innovation, technological progress, and discoveries. Unlike Barro [3] and Barro and Ursúa [4], we define a bad (or good) event as a peak-to-trough absolute decline (or a trough-to-peak absolute rise) in consumption growth rates by at least a low threshold (e.g., 3%). Thus, the events in our definition are not necessarily rare. Bad events include rare economic disasters, and good events are the opposite of bad events. There are three reasons for our definition. First, our intuitive definition means that consumption does not necessarily decline (or rise) immediately but consumption growth rates change immediately when a bad (or good) event occurs, while the Rietz-Barro model's definition of rare economic disasters implicitly assumes that consumption drops immediately when a disaster occurs. In other words, consumption growth rates are more sensitive to economic events than consumption. Second, our definition can incorporate more economic events than the Rietz-Barro model's definition under the same threshold. Take the United States as an example, as shown in Fig 1. During the 2008 international financial crisis, the consumption growth rate fell by 4.82%, while consumption fell by only 2.94%. During COVID-19, which has not yet ended, the consumption growth rate has fallen by 6.54%, while consumption has fallen by only 4.12%. Third, there is an apparent symmetry between the increasing and decreasing components in consumption growth rate curves. Thus, good events with time-varying probability can be easily incorporated into asset pricing models. As shown in Fig 1, we can observe the following characteristics of the U.S. consumption growth rate: (i) it fluctuates around a mean, (ii) the absolute decline from peak to trough has an average size of 5.48%, the average duration of 1.75 years and an average probability of 29.3% per year, and (iii) the absolute rise from trough to the peak has an average size of 5.47%, the average duration of 1.66 years and an average probability of 29.3% per year. Since there is no mean for consumption, the method of the Rietz-Barro model's definition of rare economic disasters cannot be used to define the increasing component in the consumption curve correspondingly. As a result, in the Rietz-Barro model, the drift of consumption growth rates is typically a fixed amount higher than the long-term consumption growth rate; the excess is utilized to indicate the influence of good events (e.g., institutional reform, management innovation, technological progress). This treatment is oversimplified.

Therefore, unlike the existing literature, this paper takes a redefinition of economic events to build our asset pricing model. Our definition of economic events is an extension of the Rietz-Barro model's definition of rare economic disasters. A crucial distinction between the two is that the former is based on volatility characteristics of consumption growth rates, while

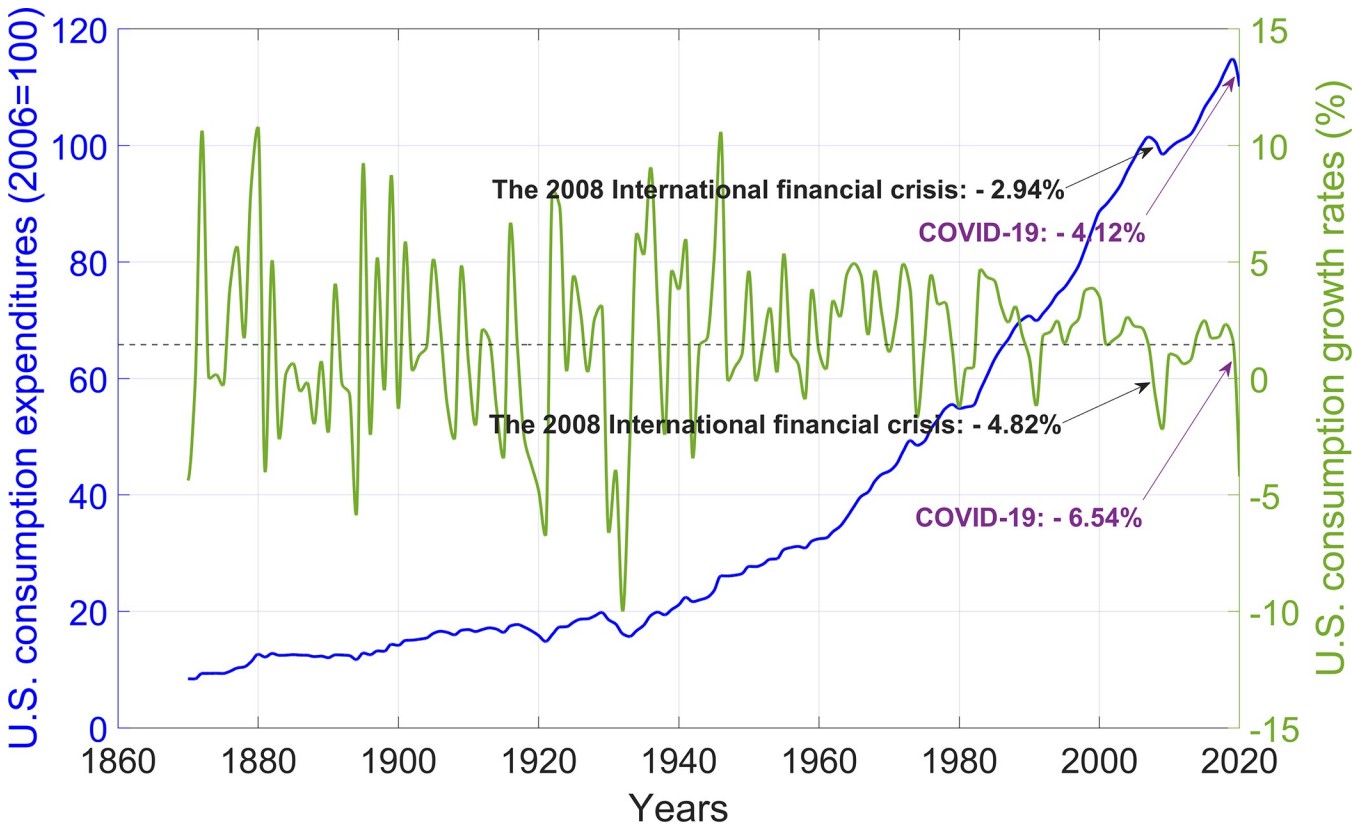

**Fig 1. U.S. consumption expenditures and U.S. consumption growth rates.** Annual data is from Barro and Ursúa [4] and updated through 2020. The dashed is the mean of U.S. consumption growth rates.

the latter is based on volatility characteristics of consumption itself. In response to the above criticisms of the Rietz-Barro model, we propose three modifications. First, model both good and bad events, not just bad events (e.g., rare economic disasters). Second, the event's impact lasts for multiple periods rather than one period. Third, model non-rare economic events.

We construct a discrete-time model in which representative agents have recursive preferences [17, 18]. The log consumption growth rate is affected by both good and bad events with time-varying probability. A good event and a bad one are modeled as a positive scale jump and a negative one, respectively. We assume that all jumps follow the Bernoulli distribution. Both good and bad events influence the log consumption growth rate through a first-order autoregressive process. Time-varying event probability is modeled by allowing the intensity of jumps to follow the square-root process [19]. Our model has a closed-form solution by taking the limiting case that sets the elasticity of intertemporal substitution to be one and making some reasonable assumptions.

The solution for the model reveals that good and bad events with time-varying probability have opposite effects on expected asset returns. Asset returns increase when a good event occurs and decrease when a bad event occurs. When their effects cancel each other out, the expected asset return in our model is equal to that in the standard model that does not contain economic events. Both good and bad events contribute positively to the variance of asset returns. The greater the volatility of event probability, the greater the volatility of asset returns. Moreover, the solution for our model implies that including good events with time-varying probability in the Rietz-Barro disaster model can further enhance the time-varying

characteristics of asset returns, but it has no significant effect on the average equity premium. The inability to predict the government bill rate volatility greater than the consumption growth rate volatility is an important challenge for the Rietz-Barro disaster model. Our model meets the challenge: it can generate both the consumption growth rate volatility observed in U.S. data (3.60%) and the government bill rate volatility observed in U.S. data (4.82%) when we set the threshold close to zero in our definition of economic events. Moreover, our model can quantitatively match the equity premium, the equity return volatility, and the Sharpe.

As we described above, our model allows for some new insights. However, it is worth noting that the simulation results of the model in this paper depend on the assumptions made about the model. There are several open questions. First question: what distribution should events be assumed to follow more reasonably? This paper assumes that events follow the Bernoulli distribution, and Wachter [7] assumes that events follow the Poisson distribution. The Poisson distribution is the limiting case of the Bernoulli distribution, so we assume that events obey the Bernoulli distribution in our discrete-time model. Second question: is it appropriate to assume the size of events to be constant? Both Wachter [7] and Barro and Jin [20] assume that the disaster size follows some probability density function. Therefore, it is necessary to endow the event size with time-varying characteristics in future research. Third question: is it reasonable to assume that the correlation between good and bad events is zero? Barro and Liao [21] show that disaster probability is highly correlated across countries. Few kinds of literature explore the correlation between good and bad events.

This paper and several recent papers on disaster recoveries share certain parallels. Good events in this paper include recoveries after disasters. While Gourio [13], Nakamura et al. [14], and Barro and Jin [15] improve the Rietz-Barro model, they still follow the Rietz-Barro model's definition of rare economic disasters. They do not model the rapidly growing component of the consumption curve unrelated to disasters.

## U.S. consumption growth rates and economic events

U.S. consumption growth rate data (1870–2020) shows 44 trough-to-peak rises and 44 peak-to-trough declines. Fig 2 shows the size distribution of the rises and declines. Generally, the number of larger rises (or declines) is less than the number of smaller rises (or declines). The average sizes of the rises and declines are 5.48% and 5.47%, respectively. Barro and Jin [20] argue that a power-law density provides a good fit to the distribution of transformed sizes of disasters defined by Barro and Ursúa [4]. However, as shown in Fig 3, it is not a very good fit for the distribution of transformed sizes of the rises and declines defined by us, especially for the distribution of transformed sizes of the rises. Fig 4 shows the duration distribution of the rises and declines. The frequency is close to a strictly decreasing function of the duration of the rises (or the declines). The average duration of the rises and declines is 1.75 years and 1.66 years, respectively.

We define a good event as a peak-to-trough absolute decline in consumption growth rates by at least $\tau^r$ ($<10\%$) and a bad event as a trough-to-peak absolute rise in consumption growth rates by at least $\tau^d$ ($<10\%$). Fig 5 shows the correspondence between the thresholds ($\tau^r$ and $\tau^d$) and the characteristics of economic events. On the whole, the average duration of bad events is longer than that of good events at different thresholds; the former is between 1.66 and 1.96, while the latter is between 1.46 and 1.83. The average cumulative effect of both good and bad events increases in the threshold. An absolute increase of 1% in the threshold brings an absolute increase of about 0.87% in the average cumulative effect of good events. In comparison, it brings an absolute decline of about 0.64% in the average cumulative effect of bad events. At different thresholds, good events and bad events have a similar average probability; an absolute increase of 1% in the threshold brings an absolute decrease of about 2.5% in the probability.

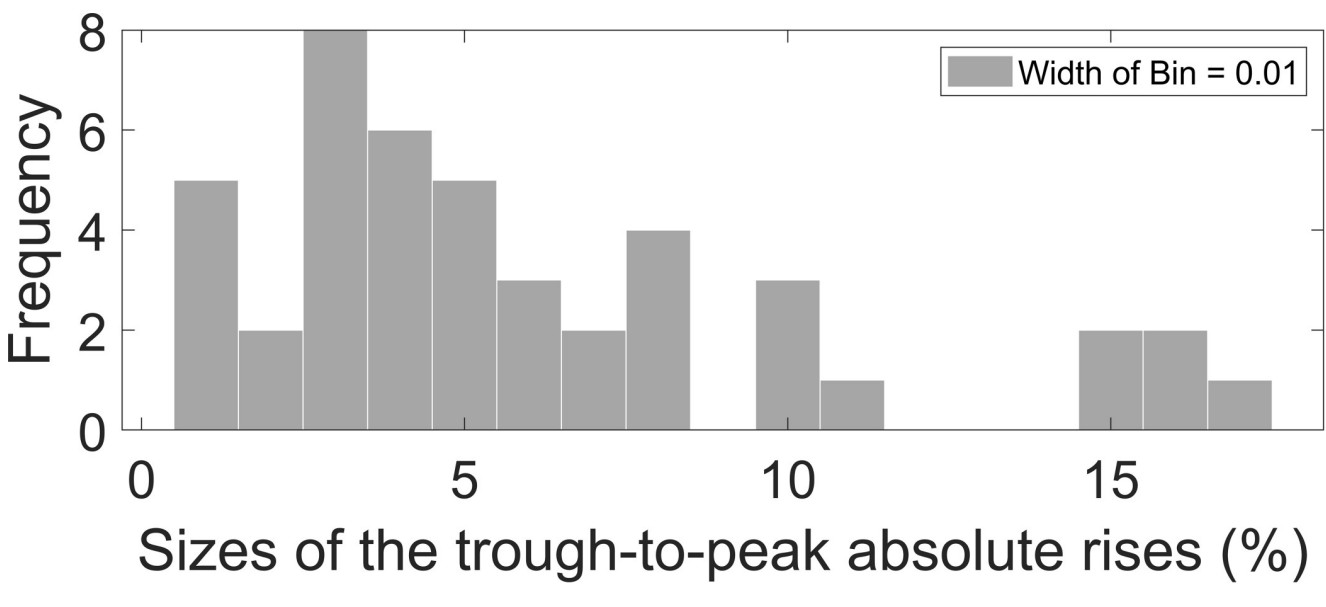

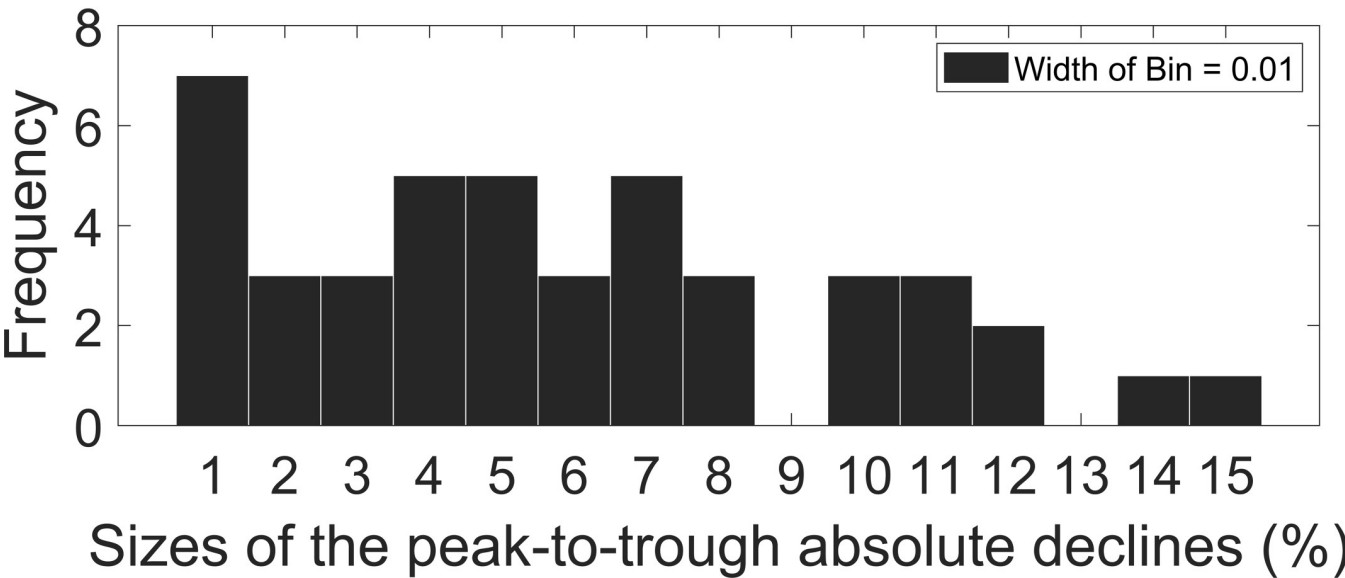

**Fig 2. Size distribution of the rises and declines in U.S. consumption growth rate data.**

### The model

We assume that agents' preferences are recursive but not time-separable [17, 18]:

$$U_t = \left[ (1-\delta)(C_t)^{1-1/\psi} + \delta E_t[(U_{t+1})^{1-\gamma}]^{\frac{1-1/\psi}{1-\gamma}} \right]^{\frac{1}{1-1/\psi}}, \tag{1}$$

where variables $C_t$ and $U_t$ represent consumption and utility of agents, respectively, $1-\delta$ is the rate of time preference ($0<\delta<1$), and $\gamma$ and $\psi$ measure relative risk aversion and elasticity of intertemporal substitution, respectively ($\gamma>1$). Unlike the power utility (CRRA), which requires $\gamma = 1/\psi$, the recursive utility allows $\gamma \neq 1/\psi$. Recursive utility implies that agents are risk-averse to future risks. In the literature, the reasonable value of $\psi$ is controversial. Hall [22] estimates $\psi$ to be close to 0. Campbell [23] and Guvenen [24] argue that $\psi$ should be less than

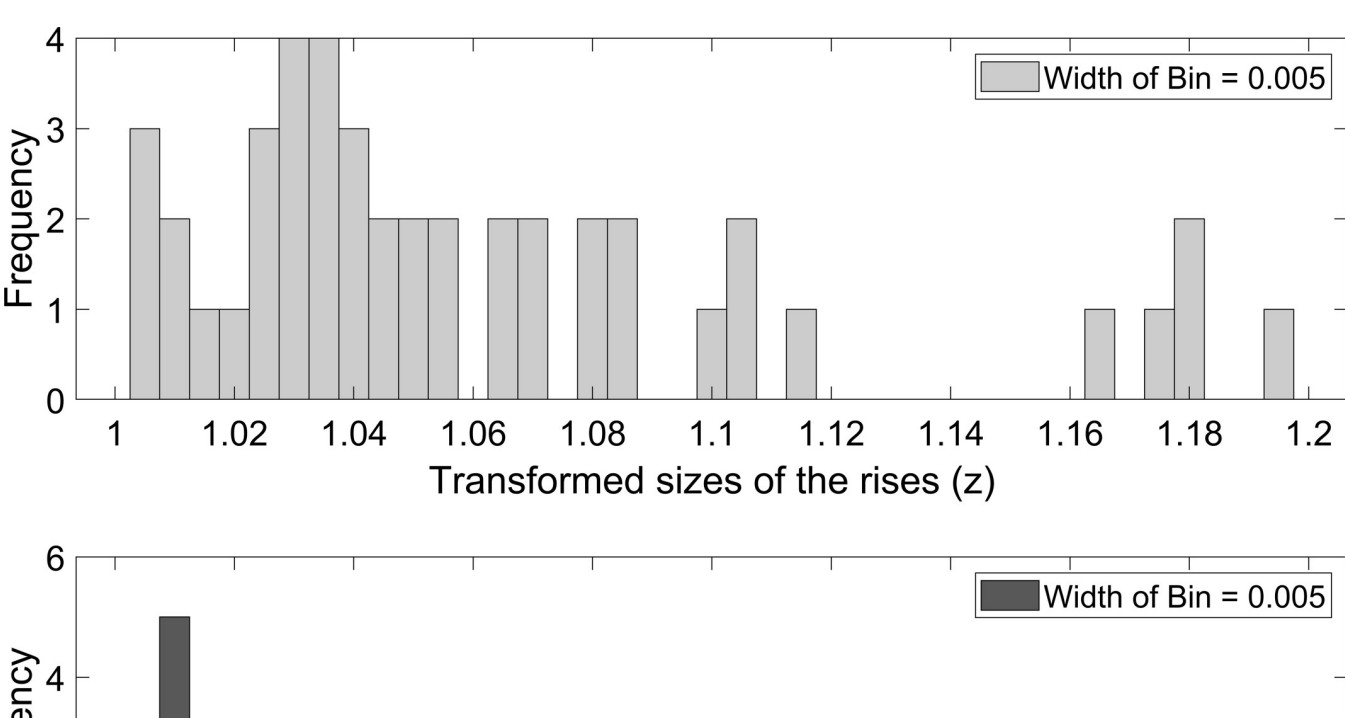

**Fig 3. The transformed size distribution of the rises and declines in U.S. consumption growth rate data.** $z = \frac{1}{1-b}$, where $z$ is the transformed size and $b$ is the size of the rises or declines.

1. Vissing-Jørgensen [25] shows that the reasonable value for $\psi$ is close to 1 or slightly lower than 1. Wachter [7] assumes that $\psi$ is equal to 1. Based on the observed behavior of asset prices during consumption disasters, Nakamura et al. [14] conclude that $\psi$ is greater than 1. Bansal and Yaron [26] set $\psi$ to be 1.5. Colacito and Croce [27] show that when $\psi$ is equal to 1.5, the model can perfectly reproduce the cross-country correlation and autocorrelation of consumption growth rates observed in the post-1970 sample. Bansal et al. [28] estimate $\psi$ to be close to 2. We set $\psi$ to 1 for tractability. Based on this, proposition 1 is given as follows.

**Proposition 1.** If we use the limiting case that sets $\psi$ in Eq (1) equal to 1, agents' preferences can change to the following form:

$$\ln U_t = (1 - \delta)\ln C_t + \frac{\delta}{1 - \gamma}\ln E_t[e^{(1-\gamma)\ln U_{t+1}}]. \tag{2}$$

See S1 Appendix for the proof of Proposition 1. It can be seen that Eq (2) still retains the key feature of Eq (1) that the relative risk aversion and the intertemporal elasticity of substitution are separated from each other. To simplify, we use Eq (2) to describe agents' preferences in the model. Agents obtain an endowment consumption stream $C_t$. We assume that the log

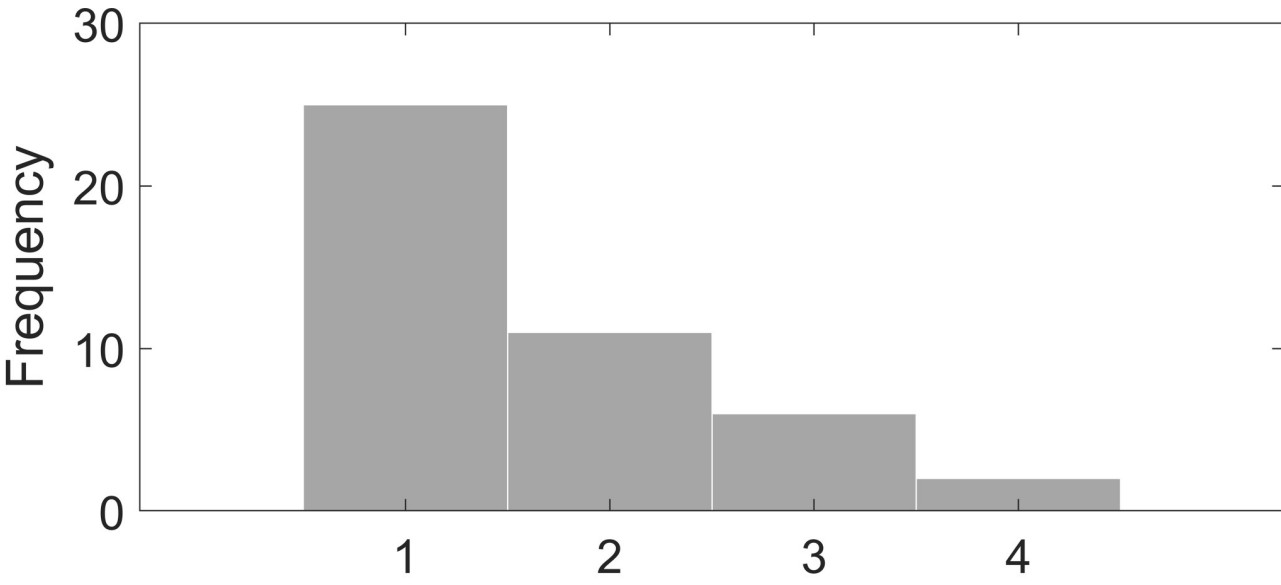

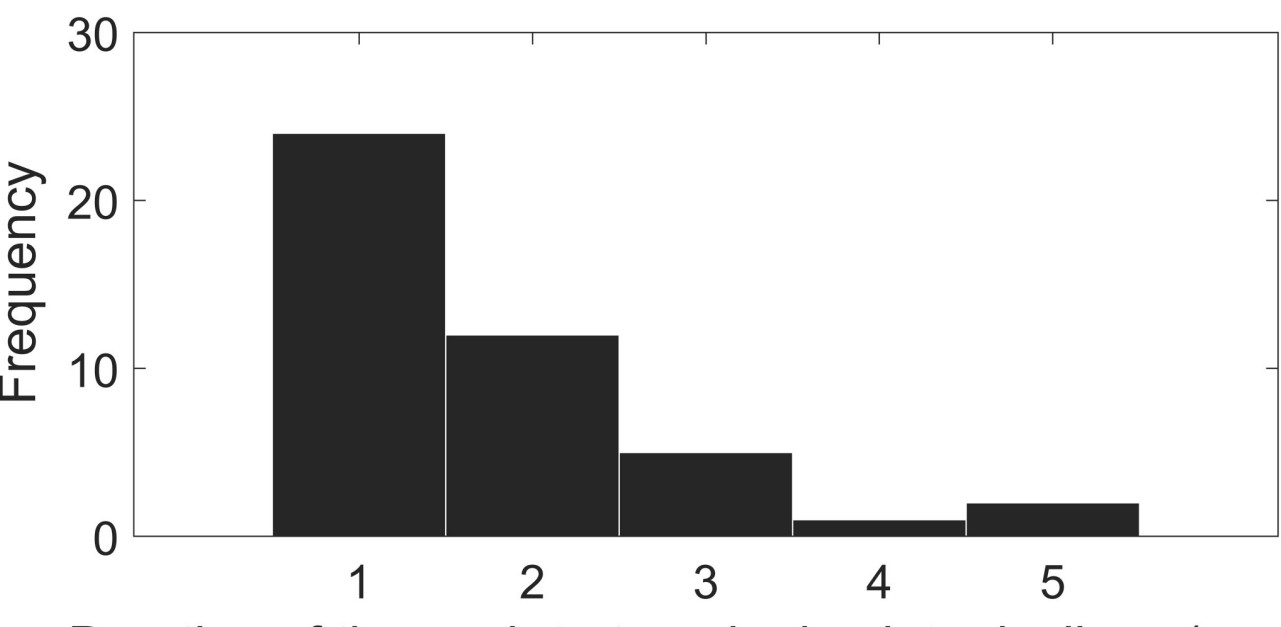

**Fig 4. Duration distribution of the rises and declines in U.S. consumption growth rate data.**

consumption growth rate is given by

$$\Delta c_{t+1} = \mu + \sigma \varepsilon_{t+1} + \eta_{1,t+1} + \eta_{2,t+1} \qquad (3)$$

where $\Delta c_{t+1} = \ln C_{t+1} - \ln C_t$, $\mu$ is the constant term, $\varepsilon_{t+1} \sim N(0,1)$, and $\sigma \varepsilon_{t+1}$ represents the short-run shock on the log consumption growth rate. In Eq (3), $\eta_{1,t+1}$ and $\eta_{2,t+1}$ are modeled as AR

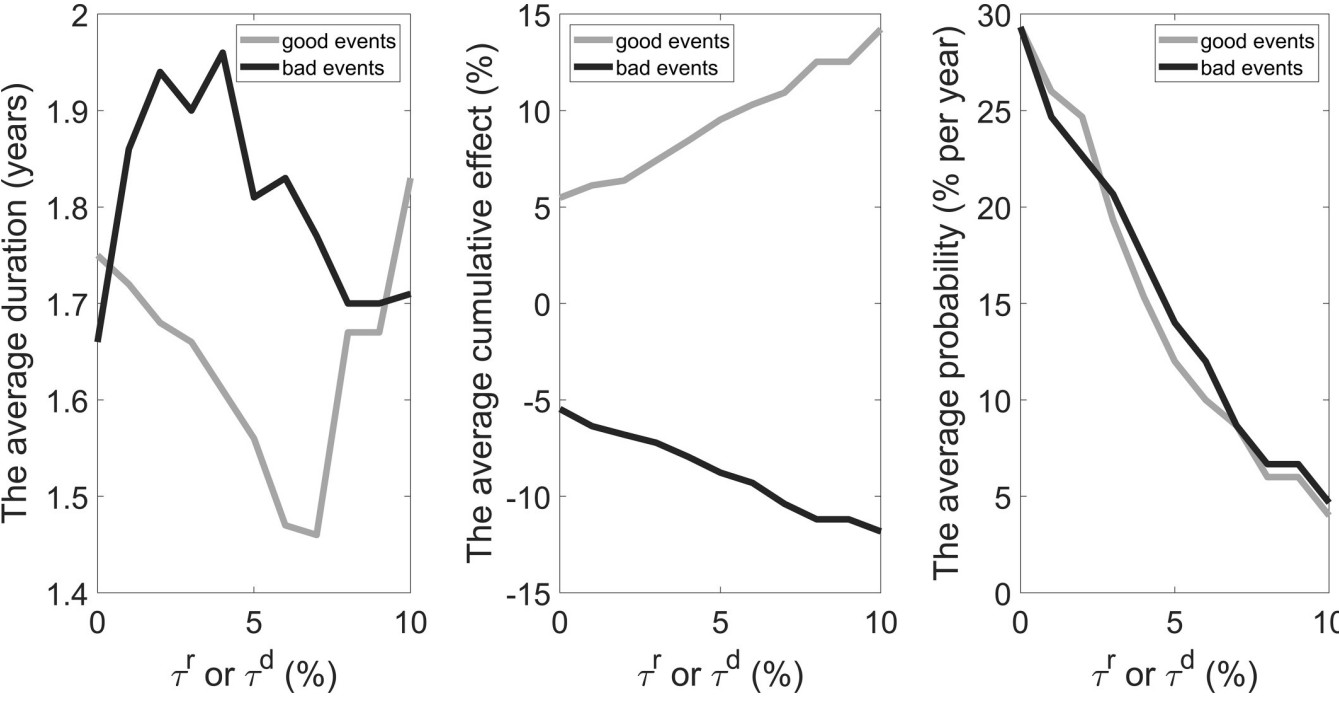

**Fig 5. Thresholds and characteristics of economic events.**

(1) processes with jumps:

$$\eta_{i,t+1} = a_i\eta_{i,t} + b_i N_{i,t+1}, i \in \{1, 2\} \tag{4}$$

where $a_i$ is the AR(1) coefficient, $N_i$ represents jumps (i.e., economic events), and $b_i$ denotes the size of jumps. Eq (4) combined with Eq (3) shows that the larger $a_i$, the more lasting the impact of events on consumption growth rates. So, it can be said that our model is somehow related to the long-run risk model [26]. $N_{i,t+1}$ follows Bernoulli distribution:

$$N_{i,t+1} = \begin{cases} 1, & \text{with prob. } \lambda_{i,t} \\ 0, & \text{with prob. } 1 - \lambda_{i,t} \end{cases}, \tag{5}$$

where $N_1 = 1$ is defined as a good event, and $N_2 = 1$ is defined as a bad event. According to our definition of economic events, the consumption growth rate increases immediately when a good event occurs, and it reduces immediately when a bad event occurs. Thus, in Eq (4), $b_1 \geq 0$ and $b_2 \leq 0$. And combining Eqs (3), (4), and (5), $a_i$ and $b_i$ together determine the cumulative effect of an economic event on consumption growth rates. In Eq (5), $\lambda_i$ represents the event probability. We assume that $\lambda_i$ obeys a square-root process:

$$\lambda_{i,t+1} = \theta_i + \phi\lambda_{i,t} + \sigma_\lambda\sqrt{\lambda_{i,t}}\varepsilon_{i,t+1}, \tag{6}$$

where $1 > \lambda_{i,t} \geq 0$ [19], $\theta_i$ is the constant term ($\theta_i > 0$), and $\phi$ determines the rate of mean reversion ($1 > \phi > 0$). The unconditional expectation of event probability is $\theta_i/(1-\phi)$. $\sigma_\lambda$ is the volatility parameter of event probability. $\varepsilon_{i,t+1} \sim N(0,1)$.

## Closed-form solutions

### The stochastic discount factor

Based on Eq (2), the log stochastic discount factor can be written as

$$m_{t+1} = \ln \delta - \Delta c_{t+1} + (1 - \gamma)\zeta_{t+1} - \ln E_t(e^{(1-\gamma)\zeta_{t+1}}) \tag{7}$$

where $\zeta_{t+1} = \ln U_{t+1} - \ln C_t$. First, assumption 1 is made to get the closed-form solution for the log stochastic discount factor.

**Assumption 1.** $\zeta_{t+1} = \varpi_{t+1} + \omega_1 N_{1,t+1} + \omega_2 N_{2,t+1}$, where $E_t(\varpi_{t+1}) = v_t$, $var_t(\varpi_{t+1}) = \chi_t^2$, $\omega_1$ and $\omega_2$ are constants, $corr(N_{1,t+1}, N_{2,t+1}) = 0$, and all other correlations are also set to zero.

**Proposition 2.** If assumption 1 holds, the following holds:

$$E_t\left(e^{(1-\gamma)\zeta_{t+1}}\right) = e^{(1-\gamma)v_t + \frac{(1-\gamma)^2}{2}\chi_t^2}\left[\lambda_{1,t}e^{(1-\gamma)\omega_1} + 1 - \lambda_{1,t}\right]\left[\lambda_{2,t}e^{(1-\gamma)\omega_2} + 1 - \lambda_{2,t}\right] \tag{8}$$

See S1 Appendix for the proof of Proposition 2. Further, assumption 2 is shown below.

**Assumption 2.** $\lambda_{i,t}e^{(1-\gamma)\omega_i} + 1 - \lambda_{i,t} \approx e^{(1-\gamma)\omega_i\lambda_{i,t}}$, where $i \in \{1,2\}$.

To verify assumption 2, we define the measurement as

$$\Gamma_i = 1 - \frac{e^{(1-\gamma)\omega_i E(\lambda_{i,t})}}{E(\lambda_{i,t})e^{(1-\gamma)\omega_i} + 1 - E(\lambda_{i,t})}, \tag{9}$$

where $i \in \{1,2\}$. If $\Gamma_i$ is close to zero, assumption 2 holds. Based on assumption 2, Eq (B1) in S1 Appendix can be written as:

$$E_t\left(e^{(1-\gamma)\zeta_{t+1}}\right) = e^{(1-\gamma)\left(v_t + \frac{1-\gamma}{2}\chi_t^2 + \omega_1\lambda_{1,t} + \omega_2\lambda_{2,t}\right)}, \tag{10}$$

**Proposition 3.** If we define $y_t = \ln U_t - \ln C_t$, then Eq (2) can be transformed into

$$y_t = \delta\left(v_t + \frac{1-\gamma}{2}\chi_t^2 + \omega_1\lambda_{1,t} + \omega_2\lambda_{2,t}\right). \tag{11}$$

See S1 Appendix for the proof of Proposition 3.

**Assumption 3.** $y_t = l_1 + l_2\eta_{1,t} + l_3\eta_{2,t} + l_4\lambda_{1,t} + l_5\lambda_{2,t}$, where $l_t$, $l_2$, $l_3$, $l_4$, and $l_5$ are unknown coefficients.

**Proposition 4.** If assumption 3 holds, then $l_1 = \frac{\delta[\mu + l_4\theta_1 + l_5\theta_2 + \sigma^2(1-\gamma)/2]}{1-\delta}$, $l_2 = \frac{\delta a_1}{1-\delta a_1}$, $l_3 = \frac{\delta a_2}{1-\delta a_2}$, $l_4 = \frac{1 - \delta\phi - \sqrt{(1-\delta\phi)^2 - 2(1-\gamma)\delta^2\sigma_\lambda^2(1+l_2)b_1}}{\delta(1-\gamma)\sigma_\lambda^2}$ and $l_4 = \frac{1 - \delta\phi - \sqrt{(1-\delta\phi)^2 - 2(1-\gamma)\delta^2\sigma_\lambda^2(1+l_3)b_2}}{\delta(1-\gamma)\sigma_\lambda^2}$.

See S1 Appendix for the proof of Proposition 4. The closed-form log stochastic discount factor is

$$m_{t+1} = \underbrace{\Theta_0 - \Theta_\varepsilon\varepsilon_{t+1}}_{standard\ model}$$

$$\underbrace{-a_1\eta_{1,t} - \Theta_{\lambda_1}\lambda_{1,t} + \Theta_{\varepsilon1}\sqrt{\lambda_{1,t}}\varepsilon_{1,t+1} + \Theta_{N_1}N_{1,t+1}}_{good\ events}$$

$$\underbrace{-a_2\eta_{2,t} - \Theta_{\lambda_2}\lambda_{2,t} + \Theta_{\varepsilon2}\sqrt{\lambda_{2,t}}\varepsilon_{2,t+1} + \Theta_{N_2}N_{2,t+1}}_{bad\ events}, \tag{12}$$

where $\Theta_0 = \ln \delta - \mu - (1-\gamma)^2 \sigma^2 / 2$, $\Theta_\varepsilon = \gamma \sigma$, $\Theta_{\lambda_1} = (1-\gamma)^2 \sigma_\lambda^2 l_4^2 / 2 + (1-\gamma)(1+l_2)b_1$, $\Theta_{\lambda_2} = (1-\gamma)^2 \sigma_\lambda^2 l_5^2 / 2 + (1-\gamma)(1+l_3)b_2$, $\Theta_{\varepsilon 1} = (1-\gamma)l_4 \sigma_\lambda$, $\Theta_{\varepsilon 2} = (1-\gamma)l_5 \sigma_\lambda$, $\Theta_{N_1} = (1-\gamma)(1+l_2)b_1 - b_1$, and $\Theta_{N_2} = (1-\gamma)(1+l_3)b_2 - b_2$.

## The risk-free rate

Assumption 4 is given first to get the closed-form solution of the log risk-free rate.

**Assumption 4.** $\lambda_{1,t}e^{\Theta_{N1}} + 1 - \lambda_{1,t} \approx e^{\Theta_{N1}\lambda_{1,t}}$ and $\lambda_{2,t}e^{\Theta_{N2}} + 1 - \lambda_{2,t} \approx e^{\Theta_{N2}\lambda_{2,t}}$.

To verify assumption 4, we define the measurement as

$$\Lambda_i = 1 - \frac{e^{\Theta_{Ni}E(\lambda_{i,t})}}{E(\lambda_{1,t})e^{\Theta_{N1}} + 1 - E(\lambda_{1,t})}, i \in \{1, 2\}. \tag{13}$$

So, the log risk-free rate satisfies

$$r_t^f = -\ln E_t(e^{m_{t+1}}) = \mu - \ln \delta + (1-2\gamma)\sigma^2/2 + a_1\eta_{1,t} + b_1\lambda_{1,t} + a_2\eta_{2,t} + b_2\lambda_{2,t}. \tag{14}$$

Further, the unconditional expectation and variance of $r_t^f$ can be obtained as follows:

$$E(r_t^f) = \underbrace{\mu - \ln \delta + (1-2\gamma)\sigma^2/2}_{\text{standard model}} + \underbrace{\frac{b_1\theta_1}{(1-a_1)(1-\phi)}}_{>0, \ good \ events} + \underbrace{\frac{b_2\theta_2}{(1-a_2)(1-\phi)}}_{<0, \ bad \ events}, \tag{15}$$

$$Var(r_t^f) = \underbrace{Var(a_1\eta_{1,t} + b_1\lambda_{1,t})}_{good \ events} + \underbrace{Var(a_2\eta_{2,t} + b_2\lambda_{2,t})}_{bad \ events}. \tag{16}$$

The term above the first bracket in Eq (15) is the same as in the standard model that does not contain economic events; $\mu$ stands for the expected consumption growth rate in normal times; $-\ln \delta$ stands for time preference; $(1-2\gamma)\sigma^2/2$ stands for the precautionary saving. The term above the second bracket in Eq (15) is derived from good events. The term above the third bracket in Eq (15) is derived from bad events.

Eq (16) implies that the risk-free rate volatility is equal to zero in the standard model, and the modeling of economic events can increase the risk-free rate volatility. Fig 6 shows the risk-free rate volatility as a function of expected event probability. The solid line represents that both good and bad events are modeled, and the dashed line represents that only bad events are modeled. In both ways of modeling, the risk-free rate volatility increases in $E(\lambda_t)$ because a higher $E(\lambda_t)$ means a higher uncertainty. The dashed line lies below the solid line—the difference between the dashed line and the solid line increases in $E(\lambda_t)$. Compared with the modeling of only bad events, adding the modeling of good events can increase risk-free rate volatility. As shown in Fig 7, the risk-free rate volatility is also a strictly increasing function of the volatility parameter of event probability.

## The equity return

We refer to Colacito and Croce [27] to assume that the log equity return ($r_{t+1}^e$) is a function of the log consumption return ($r_{t+1}^c$):

$$r_{t+1}^e = \Phi r_{t+1}^c + \sigma_d \varepsilon_{d,t+1}, \tag{17}$$

where $\Phi$ is the leverage parameter which is usually set to a number larger than one, $\sigma_d \varepsilon_{d,t+1}$ captures dividend-specific shocks, and $r_{t+1}^c$ can be expressed in terms of the log price-

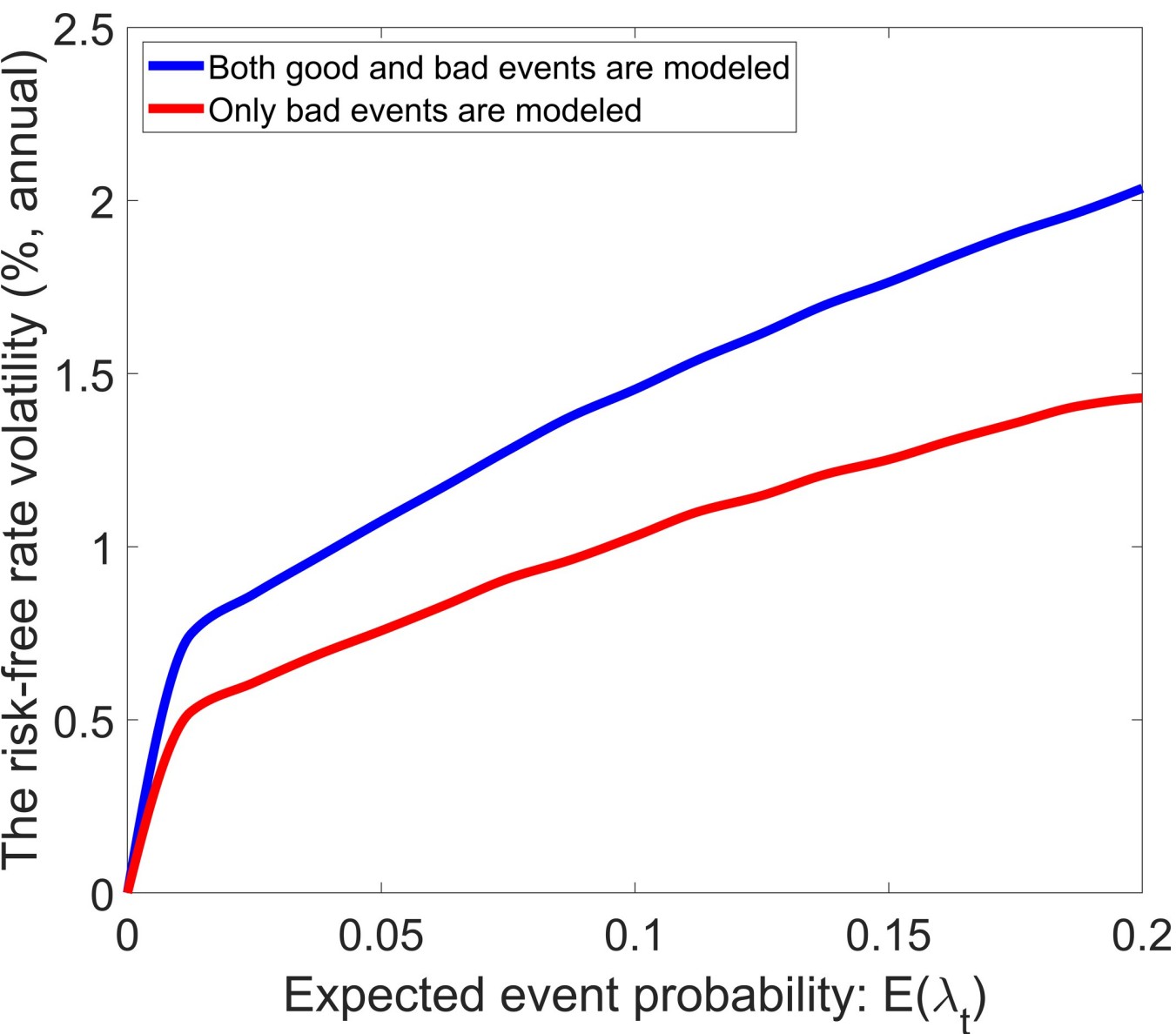

**Fig 6. Risk-free rate volatility with different values of the event probability.**

consumption ratio ($pc_t$) as

$$r_{t+1}^c = \ln(e^{pc_{t+1}} + 1) - pc_t + \Delta c_{t+1}. \tag{18}$$

First, linearizing Eq (18) around the steady state of $pc_t$ define as $pc$, we obtain:

$$r_{t+1}^c \approx \ln(e^{pc} + 1) - \ell pc + \ell pc_{t+1} - pc_t + \Delta c_{t+1}, \tag{19}$$

where $\ell = \frac{e^{pc}}{e^{pc}+1}$. Further, to obtain the closed-form solution of $r_{t+1}^c$, assumptions 5 and 6 are shown below.

**Assumption 5.** $pc_t = A_1^c + A_2^c \eta_{1,t} + A_3^c \eta_{2,t} + A_4^c \lambda_{1,t} + A_5^c \lambda_{2,t}$, where $A_1^c, A_2^c, A_3^c, A_4^c$ and $A_5^c$ are unknown coefficients.

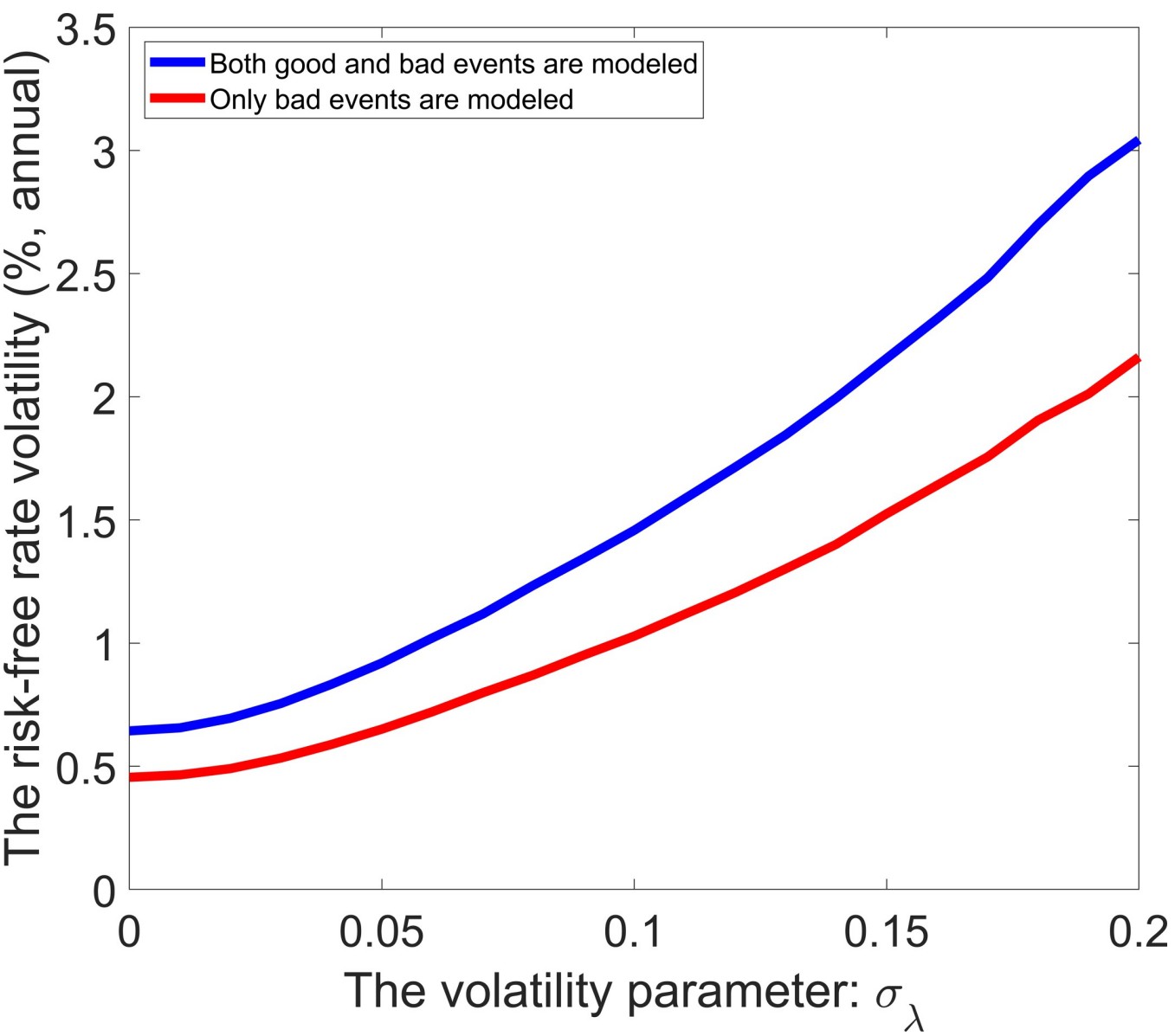

**Fig 7. Risk-free rate volatility with different values of the volatility parameter.**

**Assumption 6.** $\lambda_{1,t}e^{[\ell A_2^c + (1-\gamma)(1+l_2)]b_1} + 1 - \lambda_{1,t} \approx e^{[\ell A_2^c + (1-\gamma)(1+l_2)]b_1 \lambda_{1,t}}$ and
$\lambda_{2,t}e^{[\ell A_3^c + (1-\gamma)(1+l_3)]b_2} + 1 - \lambda_{2,t} \approx e^{[\ell A_3^c + (1-\gamma)(1+l_3)]b_2 \lambda_{2,t}}$.

To verify assumption 6, we define the measurements as

$$Y_1 = 1 - \frac{e^{[\ell A_2^c + (1-\gamma)(1+l_2)]b_1 E(\lambda_{1,t})}}{E(\lambda_{1,t})e^{[\ell A_2^c + (1-\gamma)(1+l_2)]b_1} + 1 - E(\lambda_{1,t})}, \tag{20}$$

$$Y_2 = 1 - \frac{e^{[\ell A_3^c + (1-\gamma)(1+l_3)]b_2 E(\lambda_{2,t})}}{E(\lambda_{2,t})e^{[\ell A_3^c + (1-\gamma)(1+l_3)]b_2} + 1 - E(\lambda_{2,t})}. \tag{21}$$

**Proposition 5.** If both assumption 5 and assumption 6 hold, then
$A_1^c = \frac{\ln[\delta(e^{pc}+1)]-\ell pc}{1-\ell}, A_2^c = A_3^c = A_4^c = A_5^c = 0$.

See S1 Appendix for the proof of Proposition 5. As shown in Fig 8, $pc$ can be determined by solving a fixed point problem. When $\delta$ is equal to 0.988, the steady-state value of $pc_t$ is 4.4 (i.e., $pc = 4.4$).

The closed-form log equity return is

$$r_{t+1}^e = \Phi(\mu - \ln \delta) + \Phi\sigma\varepsilon_{t+1} + \sigma_d\varepsilon_{d,t+1} + \Phi\eta_{1,t+1} + \Phi\eta_{2,t+1}. \tag{22}$$

The unconditional expectation and variance of $r_{t+1}^e$ can be obtained as follows:

$$E(r_{t+1}^e) = \underbrace{\Phi(\mu - \ln\delta)}_{\text{standard model}} + \underbrace{\frac{\Phi b_1\theta_1}{(1-a_1)(1-\phi)}}_{\text{good events}} + \underbrace{\frac{\Phi b_2\theta_2}{(1-a_2)(1-\phi)}}_{\text{bad events}}, \tag{23}$$

$$Var(r_{t+1}^e) = \underbrace{Var(\Phi\sigma\varepsilon_{t+1}) + Var(\sigma_d\varepsilon_{d,t+1})}_{\text{standard model}} + \underbrace{Var(\Phi\eta_{1,t+1})}_{\text{good events}} + \underbrace{Var(\Phi\eta_{2,t+1})}_{\text{bad events}}. \tag{24}$$

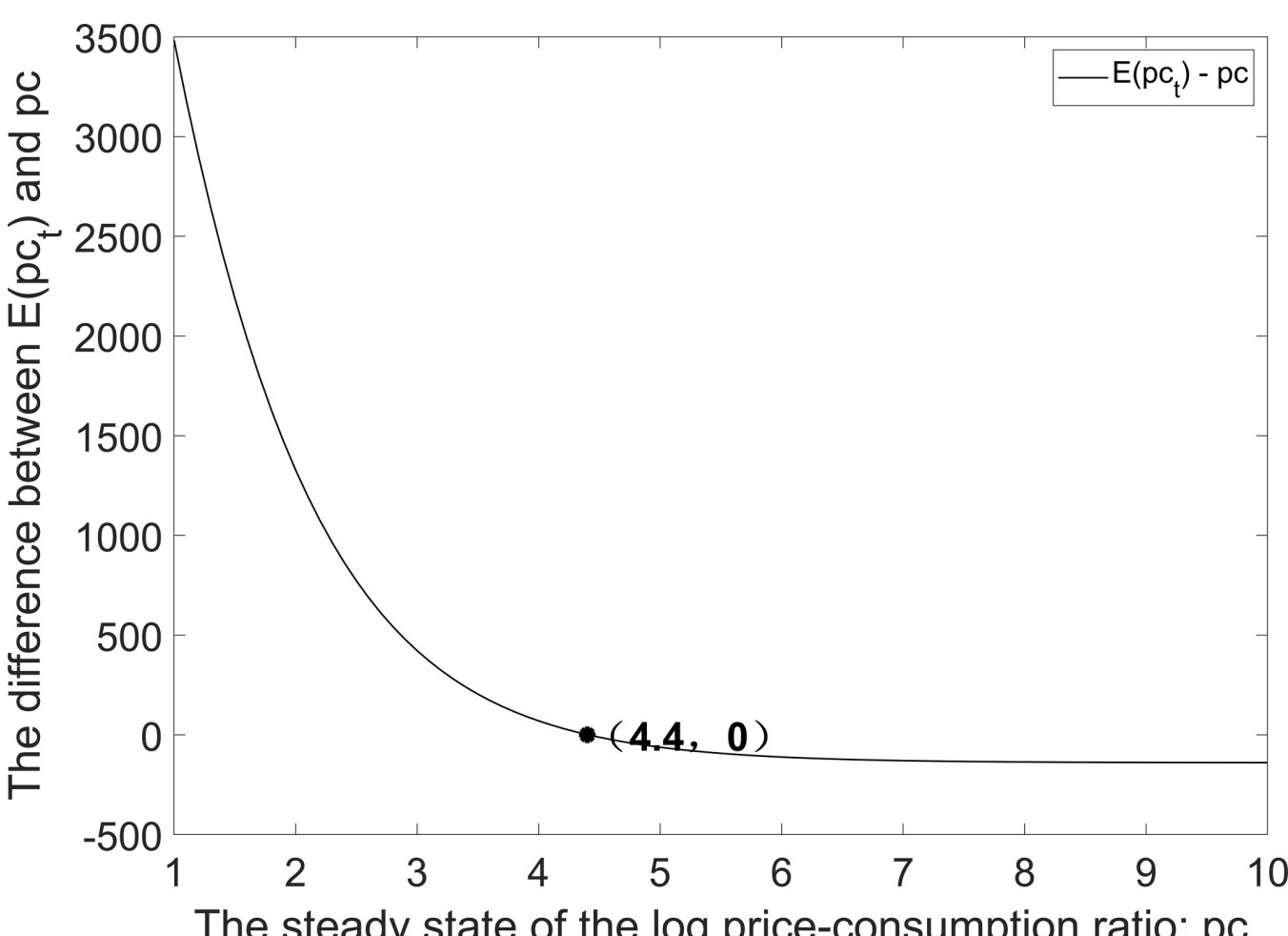

**Fig 8. Solving the fixed point problem for the steady state of log price-consumption ratios.** Set $\delta = 0.988$.

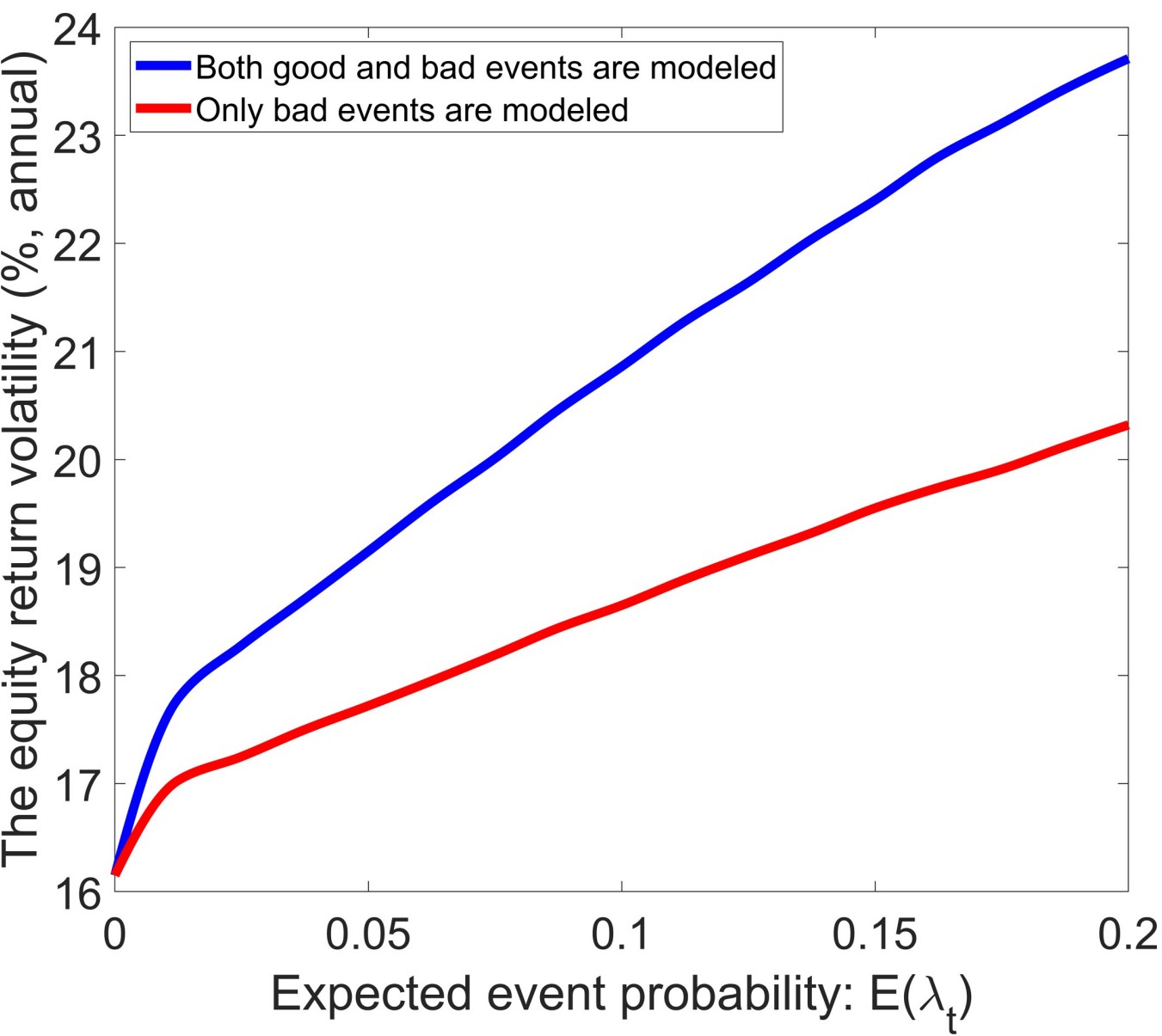

**Fig 9. Equity return volatility with different values of the event probability.**

Eq (23) indicates that both good and bad events affect the expected equity return. Fig 9 shows that the equity return volatility increases with the expected event probability. The dashed line lies below the soil line in this figure, showing that adding the modeling of good events with time-varying probability can yield a higher equity return volatility. The difference between the dashed line and the solid line increases in $E(\lambda_t)$. Fig 10 shows that the equity return volatility is also an increasing function of the volatility parameter. The greater the volatility parameter, the greater the change rate of equity return volatility. When the volatility parameter is less than a specific value, it has no significant effect on the equity return volatility.

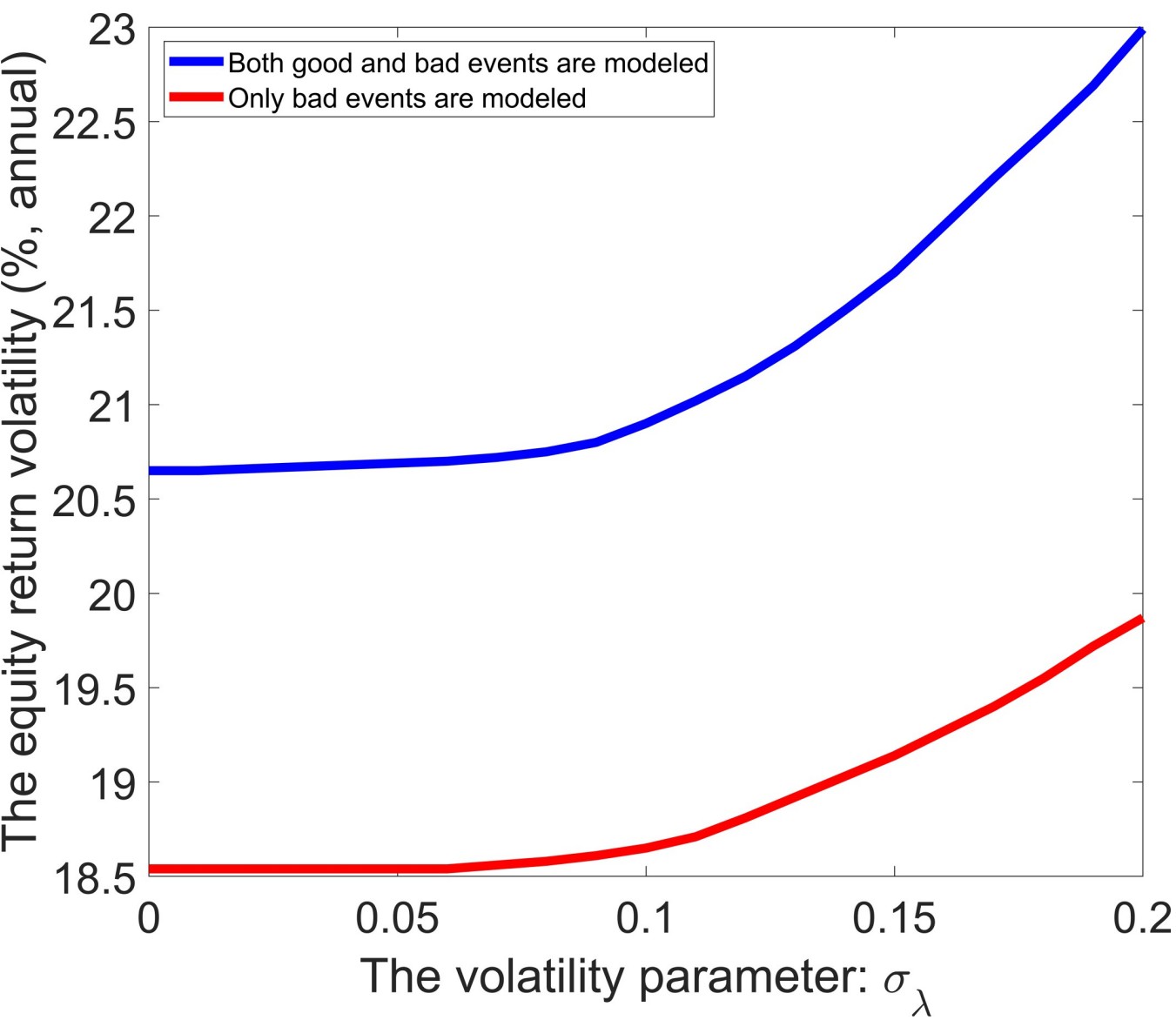

**Fig 10. Equity return volatility with different values of the volatility parameter.**

## Calibration and simulation

### Calibration

Our model measures time in years, and parameter values are given accordingly. First, we calibrate parameters independent of economic events ($\delta$, $\gamma$, $\phi$, $\Phi$, and $\sigma_d$). The rate of time preference $(1-\delta)$ is equal to 0.012 to match the average real return on the 3-month Treasury bill in postwar U.S. data [7]. The relative risk aversion ($\gamma$) is equal to 6 to follow the models allowing for faster growth following a disaster [13–15, 29]. The mean reversion parameter ($\phi$) is set to 0.92 to be consistent with the autocorrelation for the price-dividend ratio in postwar U.S. data [7]. The leverage ($\Phi$) is set to 3, which is a reasonable value by the standards of available literature [7, 16, 30, 31]. The dividend-specific volatility ($\sigma_d$) is set to 0.15 [27].

**Table 1. Descriptive statistics of economic events.**

| | Good events | | Bad events | |
|---|---|---|---|---|
| | $\tau^r = 0$ | $\tau^r = 3\%$ | $\tau^d = 0$ | $\tau^d = 3\%$ |
| Average duration (years) | 1.75 | 1.66 | 1.66 | 1.90 |
| Probability (%) | 29.33 | 19.33 | 29.33 | 20.67 |
| Average cumulative effect (%) | 5.48 | 7.41 | -5.47 | -7.22 |

Second, we discuss parameters related to economic events ($\theta_i$, $a_i$, and $b_i$). From Eq (6), the parameter $\theta_i$ determines the event probability when $\phi$ is fixed; the larger $\theta_i$, the higher the probability. We can calculate $\theta_i$ if the event probability and the parameter $\phi$ are known. The parameter $a_i$ measures the duration of an economic event, and the parameter $b_i$ represents the initial effect of an economic event. If the average duration and cumulative effect of economic events are known, $a_i$ and $b_i$ can be determined by fitting. We give specific thresholds ($\tau^r$ and $\tau^d$) to obtain the average duration, probability, and average cumulative effect of economic events. Barro and Ursúa [4] argue that raising the threshold from 10% to 15% when defining rare economic disasters will exclude many economic events but has only moderate implications for explaining asset returns. We consider two cases with thresholds ($\tau^r$ and $\tau^d$) equal to 0 and 3% to test the applicability of Barro and Ursúa's above statement to this paper. Table 1 shows descriptive statistics of good and bad events when $\tau^r = \tau^d = 0$ and $\tau^r = \tau^d = 3\%$. Fig 11 shows the fitting curves drawn according to the average duration and cumulative effect provided in Table 1. Table 2 presents parameters $a_i$ and $b_i$ back-calculated from the fitted curves in Fig 11.

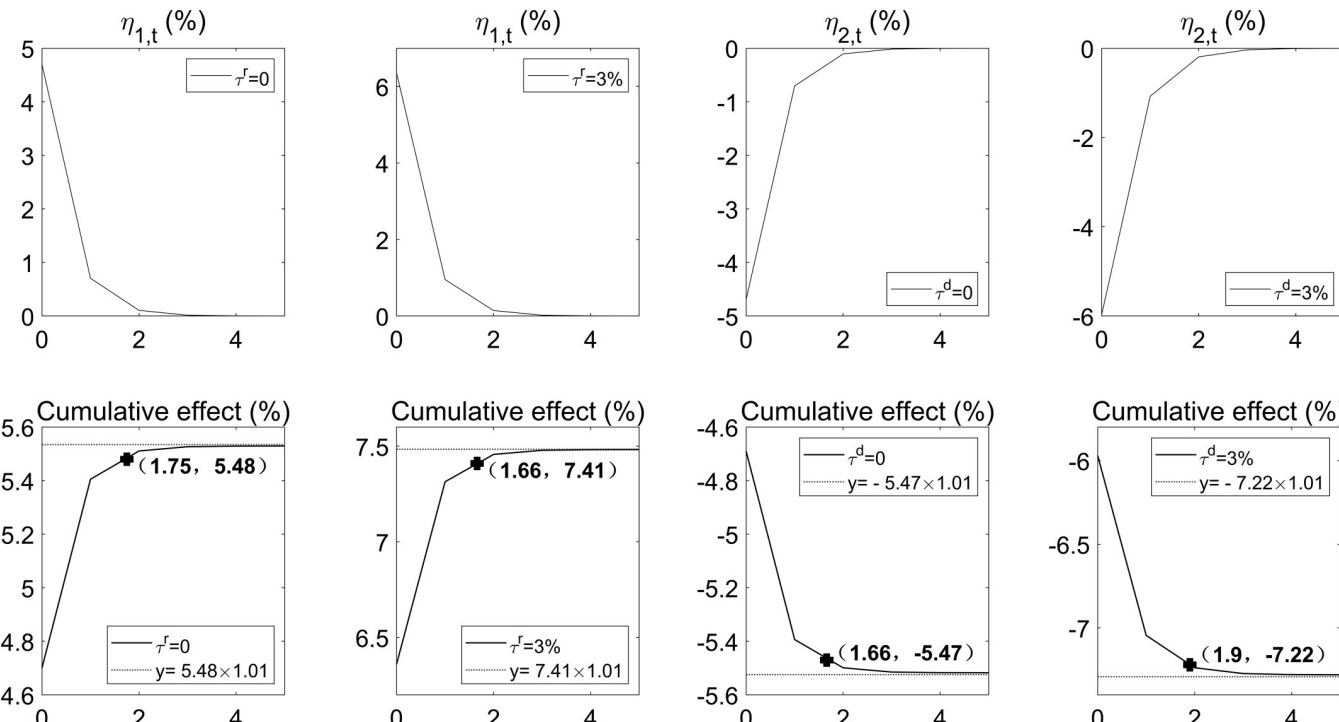

**Fig 11. Fitting curves.** The horizontal axis of all subgraphs is in years. The four subgraphs on the top show the lasting effect of a good (or bad) event occurring at $t = 0$ on the consumption growth rate. The four subgraphs on the bottom show the accumulative effect of a good (or bad) event occurring at $t = 0$ on the consumption growth rate. $\eta_{1,t} > 0$ for good events, and $\eta_{2,t} < 0$ for bad events. The fitting method is such that the cumulative curve passes through a bold coordinate point and converges to 1.01 times the average cumulative effect observed in the data. This can be seen as an approximation.

**Table 2. Event parameters for the simulation.**

| $\tau^r = \tau^d$ | $a_1$ | $a_2$ | $b_1$ | $b_2$ | $\theta_1$ | $\theta_2$ |
|---|---|---|---|---|---|---|
| 0 | 0.15 | 0.15 | 0.047 | -0.047 | 0.0235 | 0.0235 |
| 3% | 0.15 | 0.18 | 0.064 | -0.060 | 0.0155 | 0.0165 |

## Simulation results

In this subsection, we compare the simulation results of our model with those of other models to characterize our model. All models are simulated at an annual frequency for 50,000 years.

Table 3 shows the simulation results with $\tau^r = \tau^d = 0$, and it mainly consists of three parts: (i) other parameters, (ii) assumption verifications, and (iii) moments. The values of other parameters ($\mu$, $\sigma$, and $\sigma_\lambda$) are determined by matching the moments from U.S. data. Specifically, the value of the parameter $\mu$ is determined by matching the average consumption growth rate, and the values of parameters $\sigma$ and $\sigma_\lambda$ are determined by matching the consumption growth volatility. Before giving the moments of the models, we verify assumption 2 whose measurement is $\Gamma_i$, assumption 4 whose measurement is $\Lambda_i$, and assumption 6 whose measurement is $Y_i$, where $i \in \{1,2\}$. Finally, we report moments from both U.S. data and the models. The U.S. Data from Barro and Ursúa [4]. Because we assume that the probability of default on the short-term government bill in the United States is equal to zero, the short-term government bill rate is equal to the risk-free rate. In Table 3, *No* represents a model that does not include economic events, *Only_bad* represents a model that includes only bad events, and *Both* represents a model that includes both good and bad events.

Table 3 provides five specifications. For each specification, assumptions 2, 4, and 6 all hold; the maximum error is less than 1.5%. Specification 1 is our benchmark, which demonstrates the performance of a model that does not include economic events. In specification (1), $\mu$ and

**Table 3. Moments ($\tau^r = \tau^d = 0$).**

| Specifications | U.S. Data | Models | | | | |
|---|---|---|---|---|---|---|
| | | (1) | (2) | (3) | (4) | (5) |
| | | *No* | *Only_bad* | *Both* | *Only_bad* | *Both* |
| Other parameters | | | | | | |
| $\mu$ | | 0.0185 | 0.0347 | 0.0185 | 0.0423 | 0.0185 |
| $\sigma$ | | 0.036 | 0.019 | 0.019 | 0.005 | 0.005 |
| $\sigma_\lambda$ | | - | 0 | 0 | 0.38 | 0.38 |
| Assumption verifications (%) | | | | | | |
| $\Gamma_1$ | | 0 | 0.00 | 0.76 | 0.00 | 0.76 |
| $\Gamma_2$ | | 0 | 0.81 | 0.81 | 0.81 | 0.81 |
| $\Lambda_1$ | | 0 | 0.00 | 0.76 | 0.00 | 0.76 |
| $\Lambda_2$ | | 0 | 0.81 | 0.81 | 0.81 | 0.81 |
| $Y_1$ | | 0 | 0.00 | 1.03 | 0.00 | 1.03 |
| $Y_2$ | | 0 | 1.12 | 1.12 | 1.12 | 1.12 |
| Moments (Annual %) | | | | | | |
| Mean Cons Growth | 1.85 | 1.85 | 1.85 | 1.85 | 1.85 | 1.85 |
| Mean Govt Bill | 1.99 | 2.34 | 2.85 | 2.86 | 2.52 | 3.04 |
| Mean Equity Premium | 6.28 | 6.82 | 6.30 | 6.30 | 6.65 | 6.13 |
| Std Dev Cons Growth | 3.60 | 3.60 | 2.88 | 3.60 | 2.57 | 3.60 |
| Std Dev Govt Bill | 4.82 | 0.00 | 0.33 | 0.46 | 3.19 | 4.50 |
| Std Dev Equity Return | 18.66 | 18.48 | 17.30 | 18.49 | 16.87 | 18.47 |
| Sharpe Ratio | 34.13 | 36.92 | 36.42 | 34.08 | 39.45 | 33.21 |

$\sigma$ are set to 1.85% and 3.6% to match the average growth rate of consumption and the standard deviation of consumption growth. Specification 1 shows that, except for the standard deviation of government bill rates, the other moments match the data well.

Specifications (2) and (3) feature the modeling of economic events with static probability (i.e., $\sigma_\lambda = 0$ and $\lambda_i > 0$). We first set the parameter $\sigma$ in specification 3 equal to 1.9% to match the standard deviation of consumption growth. Then, we also set the parameter $\sigma$ in specification 2 to 1.9%. Therefore, comparing specifications (2) and (3) can reflect the impact of adding the modeling of good events with static probability on the standard deviation of asset returns. The standard deviation of government bill rates in specifications (2) is 0.33%, while it is 0.46% in specifications (3). Nevertheless, the standard deviation of government bill rates in specifications (3) is still much lower than that of consumption growth (3.60%). These results suggest that an asset pricing model that includes both good and bad events with static probability cannot explain the volatility puzzle that this paper focuses on.

Specifications (4) and (5) examine the modeling of economic events with time-varying probability (i.e., $\sigma_\lambda > 0$ and $\lambda_i > 0$). Specification (5) presents the performance of our model containing both good and bad events with time-varying probability, while specification (4) appears the performance of a model that includes only bad events with time-varying probability. From Eq (16), if the probability of an economic event is time-varying, the government bill rate volatility is an increasing function of $\sigma_\lambda$. Also, from Eq (3), if the probability of an economic event is time-varying, the consumption growth volatility is an increasing function of both $\sigma$ and $\sigma_\lambda$. Therefore, in specifications (4) and (5), the larger the parameter $\sigma_\lambda$ and the smaller the parameter $\sigma$, the more favorable it is to generate the government bill rate volatility higher than the consumption growth volatility. First, we determine the values of parameters $\sigma$ and $\sigma_\lambda$ in specification (5). If we assume that the value of parameter $\sigma$ in specification (5) is equal to 0.5% (a very small value), the value of parameter $\sigma_\lambda$ (0.38) in specification (5) can be inversely calculated by matching the consumption growth volatility of 3.6%. Second, for a contrast between specifications (4) and (5), we set the values of parameters $\sigma$ and $\sigma_\lambda$ in specification (4) to be the same as those in specification (5). The standard deviation of consumption growth in specifications (4) is 2.57%, while it is 3.60% in specifications (5). The standard deviation of government bill rates in specifications (4) is 3.19%, while it is 4.50% in specifications (5). These results show that adding the modeling of good events with time-varying probability can significantly increase the volatility of the government bill rate and consumption growth. While both specifications (4) and (5) present the government bill rate volatility higher than consumption growth volatility, it is clear that the moments in specification (5) match that from the U.S. data better. Overall, each of the specifications in Table 3 quantitatively matches the equity premium, the high equity return volatility, and the Sharpe ratio.

Table 4 demonstrates the simulation results with $\tau^r = \tau^d = 3\%$. We obtain three new findings by comparing the results in Tables 3 and 4. Firstly, whether the event probability is static or time-varying, economic events can bring in higher consumption growth volatility at a higher event threshold. Secondly, economic events with static probability bring higher government bill rate volatility at a higher event threshold; however, economic events with time-varying probability bring lower government bill rate volatility at a higher event threshold. Thirdly, a near-zero event threshold is a key to our theoretical model's ability to explain the volatility puzzle that this paper focuses on.

## Conclusions

In many countries (especially G7), there is a volatility puzzle: the government bill rate volatility far exceeds the consumption growth rate volatility [4, 7, 15, 16]. To address the puzzle, we

**Table 4. Moments ($\tau^r = \tau^d = 3\%$).**

| Specifications | U.S. Data | Models | | | | |
|---|---|---|---|---|---|---|
| | | (1) | (2) | (3) | (4) | (5) |
| | | *No* | *Only_bad* | *Both* | *Only_bad* | *Both* |
| Other parameters | | | | | | |
| $\mu$ | | 0.0185 | 0.0336 | 0.0190 | 0.0336 | 0.0190 |
| $\sigma$ | | 0.036 | 0.006 | 0.006 | 0.001 | 0.001 |
| $\sigma_\lambda$ | | - | 0 | 0 | 0.09 | 0.09 |
| Assumption verifications (%) | | | | | | |
| $\Gamma_1$ | | 0 | 0.00 | 0.75 | 0.00 | 1.01 |
| $\Gamma_2$ | | 0 | 1.16 | 1.16 | 1.16 | 1.16 |
| $\Lambda_1$ | | 0 | 0.00 | 0.75 | 0.00 | 1.01 |
| $\Lambda_2$ | | 0 | 1.16 | 1.16 | 1.16 | 1.16 |
| $Y_1$ | | 0 | 0.00 | 1.06 | 0.00 | 1.37 |
| $Y_2$ | | 0 | 1.59 | 1.59 | 1.59 | 1.59 |
| Moments (Annual %) | | | | | | |
| Mean Cons Growth | 1.85 | 1.85 | 1.85 | 1.85 | 1.85 | 1.85 |
| Mean Govt Bill | 1.99 | 2.34 | 3.04 | 3.04 | 3.06 | 3.05 |
| Mean Equity Premium | 6.28 | 6.82 | 6.13 | 6.13 | 6.10 | 6.08 |
| Std Dev Cons Growth | 3.60 | 3.60 | 2.54 | 3.60 | 2.50 | 3.60 |
| Std Dev Govt Bill | 4.82 | 0.00 | 0.44 | 0.59 | 0.87 | 1.21 |
| Std Dev Equity Return | 18.66 | 18.48 | 16.83 | 18.49 | 16.76 | 18.50 |
| Sharpe Ratio | 34.13 | 36.92 | 36.44 | 33.15 | 36.41 | 32.89 |

extend the traditional definition of rare economic disasters and propose an asset pricing model that contains both good and bad events with time-varying probability. Compared with traditional disaster models, our model contains three modifications: (i) model good and bad events, not just bad ones (e.g., rare economic disasters); (ii) the event's impact lasts for multiple periods rather than one period; (iii) model non-rare economic events. In our model, a near-zero event threshold is key to bringing the government bill rate volatility higher than the consumption growth volatility. Our modeling of both good and bad events implies that the consumption growth rate is in the tail with some probability, not just in the left tail. We argue that the inclusion of good events with time-varying probability in the traditional rare disaster model is an essential complement to understanding asset return volatility, especially the government bill rate volatility.

## Supporting information

**S1 Data.**
(XLSX)

**S1 Appendix.**
(DOCX)

## Acknowledgments

We thank Ke Du for assistance in solving the model.

## Author Contributions

**Conceptualization:** Chao Xiao.

**Data curation:** Yuan Zhao, Yikang Tian.

**Formal analysis:** Chao Xiao, Yuan Zhao, Yikang Tian.

**Methodology:** Chao Xiao, Yu Lou.

**Project administration:** Chao Xiao.

**Software:** Chao Xiao.

**Supervision:** Yu Lou, Jie Liu.

**Validation:** Yu Lou.

**Visualization:** Chao Xiao.

**Writing – original draft:** Chao Xiao, Yu Lou, Jie Liu.

**Writing – review & editing:** Chao Xiao, Yu Lou.

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
