## [Decision Letter · Decision Letter 0]

21 Sep 2022

PONE-D-22-12462Can an asset pricing model containing both good and bad events with time-varying probability predict the government bill rate volatility?PLOS ONE

Dear Dr. LOU,

Thank you for submitting your manuscript to PLOS ONE. After careful consideration, we feel that it has merit but does not fully meet PLOS ONE’s publication criteria as it currently stands. Therefore, we invite you to submit a revised version of the manuscript that addresses the points raised during the review process.

We look forward to receiving your revised manuscript.

Kind regards,

Aurelio F. Bariviera, Ph.D.

Academic Editor

PLOS ONE

Reviewers' comments:

Reviewer's Responses to Questions

**Comments to the Author**

1. Is the manuscript technically sound, and do the data support the conclusions?

Reviewer #1: Yes

Reviewer #2: Yes

2. Has the statistical analysis been performed appropriately and rigorously? 

Reviewer #1: Yes

Reviewer #2: Yes

3. Have the authors made all data underlying the findings in their manuscript fully available?

Reviewer #1: Yes

Reviewer #2: Yes

4. Is the manuscript presented in an intelligible fashion and written in standard English?

Reviewer #1: No

Reviewer #2: Yes

5. Review Comments to the Author

Reviewer #1: Comments:

1. Section simulation: A simulation study is provided to demonstrate the proposed methodology. Clearly discuss the choices of the stated models and the parameters used.

2. The authors have provided some propositions and each the proof is provided. I would suggest the authors include those important proofs in the appendix such as proposition 1.

3. Figure 3: (i) What is the value of b? (ii) Explain the claim “Lines 151-152: it is not a very good fit to the distribution of transformed sizes of the series” and provide statistical evidence to justify your claims. (iii) label the x- and y-axes. (iv) How to obtain the equations y=40*z^-30 and y=15*z^-20.

4. Line 159: Figure 5 provides different plots and not descriptive statistics. Explain what is meant by “Figure 5 shows descriptive statistics of events with different event thresholds.”.

5. There are some typos and spelling errors to be corrected such as line 49: “Dad”, line 111: “Our model”. The authors should rewrite and check the paper thoroughly.

6. Line 30: “Barron and Jin [13]”. Incorrect citation.

7. Abstract: Rephase some sentences such as “This paper extended…” and “In this paper, …”.

8. Line 198: Replace “.” with “,” after “where delta_c, mu, epsilon and etc”. Similar comment applied for other section.

9. Some papers in the list of references with incomplete information and incorrect formatting.

10. Proofreading is also required

Reviewer #2: This article seems to be a significant contribution to the understanding of the volatility puzzle of the Rietz-Barro model. The puzzle is that the classical model predicts that the government bill rate volatility is smaller than the consumption ones. The model developed in this paper is able to predict higher government volatility, which in itself helps shedding some lights on this puzzle.

Minor comments

1) Check this introduction, p. 4, lines 95 and 110, there are capital letters missing.

2) Please see if Cochrane (2005) provides a relevant discussion on the equity premium puzzle.

Reference

Cochrane, J.H. (2005). Asset pricing. Revised edition. Princeton University Press.

6. PLOS authors have the option to publish the peer review history of their article (what does this mean?). If published, this will include your full peer review and any attached files.

Reviewer #1: No

Reviewer #2: No

---

## [Author Response · Author response to Decision Letter 0]

30 Sep 2022

In the Rebuttal letter, we have provided point‐by‐point responses to the comments raised by the reviewers. We have prepared the Manuscript according to journal requirements.

---

## [Decision Letter · Decision Letter 1]

5 Oct 2022

Economic events and the volatility of government bill rates

PONE-D-22-12462R1

Dear Dr. LOU,

We’re pleased to inform you that your manuscript has been judged scientifically suitable for publication and will be formally accepted for publication once it meets all outstanding technical requirements.

Kind regards,

Aurelio F. Bariviera, Ph.D.

Academic Editor

PLOS ONE

Additional Editor Comments (optional):

Reviewers' comments:

Reviewer's Responses to Questions

**Comments to the Author**

1. If the authors have adequately addressed your comments raised in a previous round of review and you feel that this manuscript is now acceptable for publication, you may indicate that here to bypass the “Comments to the Author” section, enter your conflict of interest statement in the “Confidential to Editor” section, and submit your "Accept" recommendation.

Reviewer #2: All comments have been addressed

2. Is the manuscript technically sound, and do the data support the conclusions?

Reviewer #2: (No Response)

3. Has the statistical analysis been performed appropriately and rigorously? 

Reviewer #2: (No Response)

4. Have the authors made all data underlying the findings in their manuscript fully available?

Reviewer #2: (No Response)

5. Is the manuscript presented in an intelligible fashion and written in standard English?

Reviewer #2: (No Response)

6. Review Comments to the Author

Reviewer #2: (No Response)

7. PLOS authors have the option to publish the peer review history of their article (what does this mean?). If published, this will include your full peer review and any attached files.

Reviewer #2: No

---

## [Editor Report · Acceptance letter]

10 Oct 2022

PONE-D-22-12462R1 

Economic events and the volatility of government bill rates 

Dear Dr. Lou:

I'm pleased to inform you that your manuscript has been deemed suitable for publication in PLOS ONE. Congratulations! Your manuscript is now with our production department. 

Kind regards, 

on behalf of

Dr. Aurelio F. Bariviera 

Academic Editor

PLOS ONE